# Preparation, Antioxidant and Immunoregulatory Activities of a Macromolecular Glycoprotein from *Salvia miltiorrhiza*

**DOI:** 10.3390/foods11050705

**Published:** 2022-02-27

**Authors:** Hai-Yu Ji, Ke-Yao Dai, Chao Liu, Juan Yu, Xiao-Yu Jia, An-Jun Liu

**Affiliations:** 1College of Food Science and Engineering, Tianjin University of Science and Technology, Tianjin 300457, China; haiyu11456@tust.edu.cn (H.-Y.J.); 19847910@mail.tust.edu.cn (K.-Y.D.); 21942016@mail.tust.edu.cn (C.L.); yujuan14615@tust.edu.cn (J.Y.); laj@tust.edu.cn (A.-J.L.); 2Tianjin Key Laboratory of Postharvest Physiology and Storage of Agricultural Products, Key Laboratory of Storage of Agricultural Products, National Engineering Technology Research Center for Preservation of Agricultural Products, Ministry of Agriculture and Rural Affairs, Tianjin 300384, China; 3State Key Laboratory of Food Nutrition and Safety, Tianjin University of Science and Technology, Tianjin 300457, China

**Keywords:** *S. miltiorrhiza* glycoprotein, optimization, structural properties, bioactivities

## Abstract

*Salvia miltiorrhiza* has exhibited various bioactive functions due to the existence of polysaccharides, hydrophilic phenolic acids, diterpenoid quinones, and essential oils. However, little research has reported the glycoprotein preparation and corresponding bioactivities. In this study, the water-soluble glycoprotein from *S. miltiorrhiza* roots was firstly isolated with the extraction process optimized by response surface methodology, and then, the preliminary structural properties, and the antioxidant and immunoregulatory activities were investigated. Results showed that the extraction conditions for higher extraction yields were identified as follows: ultrasonic power of 220 W, ultrasonic time of 2.0 h, extraction temperature of 60 °C, liquid/solid ratio of 20 mL/g, and the glycoprotein yields of 1.63 ± 0.04%. Structural analysis showed that the glycoprotein comprised protein and polysaccharide (contents of 76.96% and 20.62%, respectively), with an average molecular weight of 1.55 × 10^5^ Da. Besides, bioactivities analysis showed that the glycoprotein presented strong scavenging effects on multiple free radicals, and effectively enhanced the antioxidant enzyme activities and immunological indicators in cyclophosphamide-induced immunocompromised mice dose-dependently. These data demonstrated that *S. miltiorrhiza* glycoprotein presented the potential to be a novel edible functional compound, and could be practically applied in the food industry.

## 1. Introduction

*Salvia miltiorrhiza* (named Danshen in Chinese) has been used for treating cerebrovascular and cardiovascular diseases through promoting blood circulation and removing blood stasis in vivo [1,2]. The primary bioactive materials in *S. miltiorrhiza* mainly comprise polysaccharides, hydrophilic phenolic acids, diterpenoid quinones, and essential oils, which contribute to the various bioactivities, including antioxidant, anticoagulation, anti-hyperlipidemia, etc. [3,4,5]. As reported, distilled water is generally used for protein and polysaccharide extraction from multiple raw materials, and various auxiliary conditions, including ultrasonic wave and high temperature, are commonly used for improving the extraction efficiency, and various bioactive compounds prepared under different technologies always present varying degrees of activities [6,7,8]. Response surface methodology (RSM) can evaluate the impacts of processing parameters on response values by planning experiments and constructing models [9], and has been applied for optimizing the extraction processes of bioactive compounds [10]. Therefore, in this study, the *S. miltiorrhiza* glycoprotein (SMGP) was prepared under ultrasonic extraction, and the extraction conditions were optimized by RSM, which would be of benefit to its efficient utilization and use in industrial production.

Glycoproteins are a kind of proteins branched with moiety carbohydrate, which can be mainly found in plant cells. The subcellular localization (secreted or membrane-associated) and the extent of glycosylation present crucial effects on protein structure and relevant bioactivities [11]. In glycoproteins, carbohydrates are responsible for the selective interaction, and are involved in regulating many cellular processes [12]. The traditional extraction method for water-soluble bioactive compounds is generally water extraction, and then, alcohol solution is employed for the precipitation [13]. As is known, *S. miltiorrhiza* was usually used as the main component of medicinal liquor (ethanol volume fraction of 50~60%) in China, and the ingredients prepared by traditional methods might present great differences with the bioactive substances dissolved in medicinal liquor [14]. Therefore, the bioactive glycoprotein from *S. miltiorrhiza* 60% was extracted using ethyl alcohol solution as the solvent in this study, aiming to provide a data foundation for the further application of medicinal liquor containing *S. miltiorrhiza*.

Immune responses, including immune defense, regulation, and surveillance, in vivo demonstrate important roles in protecting humans from various diseases [15,16]. As reported, multiple immune cells and their secretions can exert defensive and protective effects on body through efficaciously eliminating foreign pathogens [17,18]. Reactive oxygen species (ROS) can maintain the balance of the immune system, whereas the overproduction can damage the deoxyribonucleic acid and proteins of cells, which induces the incidence of various diseases, including immunosuppression, inflammatory, cardiovascular diseases, and cancer [19,20]. As reported, the number of immunocompromised patients has steadily increased in recent years, and quality-of-life has been severely affected [21]. Therefore, the development of novel bioactive compounds on immunization and antioxidant level improvement of the body is urgent. Cyclophosphamide (CTX) treatment can reduce the antioxidant ability, promote lipid peroxidation, and induce the immunosuppression of animals in vivo, and has been widely applied for constructing immune-deficient mice models [22]. Thus, in this study, the immunosuppressed mice model via CTX induction was established for the antioxidant and immunoregulation activity evaluation of SMGP.

In the present study, the glycoprotein (SMGP) was firstly isolated from the root powder of *S. miltiorrhiza* using 60% ethanol solution as the solvent, and the optimal extraction conditions were determined by RSM. Subsequently, the preliminary structural characteristics were identified via analyzing the molecular weight and chemical components, and then, the antioxidant and immunoregulation activities of SMGP in vivo were further evaluated through establishing a CTX-induced immune-deficient mice model. These results can provide a new idea for water-soluble active substance extraction, and can promote the practical applications of SMGP as an edible functional compound in the food industry, which could also help to improve the nutritional values of medicinal liquor containing *S. miltiorrhiza*.

## 2. Materials and Methods

### 2.1. Materials and Regents

The dried *S. miltiorrhiza* were provided by Tianjin Taijin Technology Co., Ltd. (Tianjin, China). The antioxidant capacity assay kits (ABTS, DPPH, OH^−^, O_2_^−^), antioxidant enzyme activities detection kits (SOD and GSH-PX), malondialdehyde (MDA) assay kit, immunoglobulin G assay kit, immunoglobulin M assay kit, and cytokine detection kits (TNF-α, IFN-γ, IL-2, IL-4) were bought from Nanjing Jiancheng Bioengineering Institute (Nanjing, China), All of the other regents were of analytical grade.

### 2.2. Preparation of SMGP

The dried root powder of *S. miltiorrhiza* was extracted using 60% ethanol solution in an airtight container under ultrasonic assistance (Ultrasonic Cell Crusher JY92-IIDN from Ningbo Scientz Biotechnology Co., Ltd.) (Ningbo, China). Subsequently, the supernatant was obtained and concentrated, and the final ethanol volume of 80% was employed for the substances’ precipitation. Then, these sediments were dissolved in distilled water, and purified through a water dialysis method (MWCO of 100,000 Da, dialysis time of 48 h). Finally, the SMGP was obtained after lyophilization. The yield of SMGP was calculated as Formula (1):Yield of SMGP (%) = (SMGP weights/*S. miltiorrhiza* roots powder weights) × 100(1)

### 2.3. Optimization of SMGP Extraction Process

The effects of four factors, including ultrasonic power (*A:* 150, 200, 250 W), ultrasonic time (*B:* 1.5, 2.0, 2.5 h), liquid/solid ratio (*C:* 15, 20, 25 mL/g), and extraction temperature (*D:* 50, 60, 70 °C), on SMGP extraction were evaluated by Box–Behnken design (BBD). These twenty-nine combinations at three levels were executed, and each combination was determined in triplicate.

### 2.4. Chemical Components Determination of SMGP

The phenol/sulfuric acid method [23], Coomassie brilliant blue method [24], and carbazole/sulfuric method [25] were used to detect the polysaccharide, protein, and uronic acid contents of SMGP, respectively.

### 2.5. Ultraviolet Scanning and Molecular Weight Detection of SMGP

The chemical constitution of SMGP was also confirmed by a 2500PC UV-Vis spectrophotometer (Shimadzu, Japan) through the ultraviolet full wavelength (200 nm~600 nm).

The average molecular weight of SMGP was evaluated using high performance gel permeation chromatography (HPGPC) equipped with a TSK-gel G4000PWxL (7.8 mm × 300 mm), according to a previous reported method [26], and T-series dextrans of T110, T70, T40, T10, and T3 were employed as standards.

### 2.6. Monosaccharides and Amino Acids Compositions of SMGP

The monosaccharide constituents of SMGP were determined using gas chromatography (GCMS7890B-7000C, Agilent, CA, America), equipped with an HP-5 column, according to previous method after slight modifications [27]. Six monosaccharides, including L-rhamnose, D-arabinose, D-xylose, D-mannose, D-glucose, and D-galactose, were employed as standards. The amino acid compositions of SMGP after acid hydrolysis (hydrochloric acid concentration of 8 mol/L, hydrolysis temperature of 114 °C, hydrolysis time of 10 h) were determined using an amino acid automatic analyzer (L-8900, Japan Hitachi Corp, Tokyo, Japan).

### 2.7. Antioxidant Activity In Vitro

The antioxidant properties of SMGP on ABTS, DPPH, OH^−^, and O_2_^−^ free radicals were determined according to the instructions. The SMGP concentrations of 0.25, 0.5, 1, 2, and 3 mg/mL were chosen for scavenging effects evaluation, and the same concentrations of vitamin C (Vc) were used as positive control.

### 2.8. Animal Experiments Design

Fifty female Kunming mice (8 weeks) were purchased from SPF (Beijing) Biotechnology Co., Ltd. (Beijing, China) with a production license number of SYXK(Jing)2019–0010, and raised in the experimental animal room with a license number of SYXK(Jin)2018-0001 with relative humidity of 45~55%), and a controllable temperature of 20~25 °C. All animal-related experiments were conducted in accordance with the principles of Laboratory Animal Care, and approved by the Local Ethics Committee for Animal Care and Use at Tianjin University of Science and Technology.

After an acclimatization of 7 d, all mice were divided into five groups at random, with 10 mice per group: blank group, model group, and SMGP groups (50 mg/kg, 100 mg/kg, and 150 mg/kg). Firstly, the normal saline solution of 0.2 mL was employed for intragastrical administration in the blank and model groups every day; meanwhile, corresponding concentrations of SMGP were orally administrated in SMGP groups. After 7 days, these mice, except the blank group, were intra-peritoneally injected with cyclophosphamide (CTX, 30 mg/kg) for 15 d, and the gastric perfusions were simultaneously conducted. Subsequently, all mice were sacrificed, and their thymuses were collected and weighed, and the thymus indices were calculated as the ratios of the weights (mg) to body weights (g).

### 2.9. Antioxidant Effects In Vivo

The in vivo antioxidant activities of SMGP were evaluated via detecting antioxidant enzyme activities and MDA contents using the corresponding kits.

### 2.10. Immune Cells Activities Detection

The NK (natural killer) cells, splenic lymphocytes, and peritoneal macrophages activities of mice in each group were determined referring to previous method [26].

### 2.11. Antibodies and Cytokines Determination

The antibody (IgG, IgM) and cytokine (IL-2, IL-4, TNF-α, and IFN-γ) expressions in sera were evaluated by the corresponding ELISA kits following the instructions.

### 2.12. Statistical Analysis

All values in this study were expressed as the mean ± standard deviation (S.D.), and each experiment was detected in triplicate. The significance of between-group differences was determined by Student’s *t*-test and one-way analysis of variance (ANOVA): the value of *p* < 0.05 was deemed as significant.

## 3. Results and Discussions

### 3.1. Optimization of SMGP Extraction

The ultrasonic power of 200 W, ultrasonic time of 2 h, liquid/solid ratio of 20 mL/g, and extraction temperature of 60 °C were chosen as intermediate values in the BBD experiment. Table 1 shows the average actual/predicted SMGP yields, and the results demonstrate that the actual SMGP yields were close to the corresponding predicted values, suggesting a high-precision fitted model [10]. The final equation according to four coded factors is as following (2):Yields = 1.61 + 0.10 × *A −* 0.06 × *B* + 0.05 × *C* + 0.14 × *D −* 0.02 × *AB* + 0.01 × *AC* + 0.09 × *AD −* 0.05 × *BC −* 0.07 × *BD* + 0.01 × *CD −* 0.16 × *A*^2^
*−* 0.18 × *B*^2^
*−* 0.23 × *C*^2^
*−* 0.20 × *D*^2^(2)

The multi-regression model ANOVA in BBD was employed to measure the interaction effects of these factors on SMGP yields, and the results are presented in Table 2. As displayed, the *F* value of 39.34 and *p* value of <0.0001 indicated a highly significant model. The *p* value of lack of fit was 0.7268, which also suggested a fitted model [28]. Besides, statistical indicators, including *R^2^*, Adj *R^2^*, Pred *R^2^*, and CV values, were 0.9752, 0.9504, 0.8935, and 3.18, respectively, and the difference of Pred *R^2^* and Adj *R^2^* was less than 0.2 (0.0569), indicating a good correlation between actual and predicted values [29]. The parameters, including *A*, *B*, *C*, *D*, *AD*, *BC*, *BD*, *A*^2^, *B*^2^, *C*^2^, and *D*^2^, exhibited remarkable differences (*p* < 0.05) to SMGP yields, which was a similar tendency observed in a previously reported paper [30].

### 3.2. Interactive Effects Analysis

The two-dimensional (2D) and three-dimensional (3D) maps presenting the interaction effects between every two factors on SMGP yields are displayed in Figure 1. As presented, these response surface curves exhibited a maximum point in the experimental ranges, suggesting a reasonable factors’ ranges selection. The elliptical/circular contour plots could reflect the significant/indistinctive interaction effects between these variables [31]. Therefore, Figure 1C–E,c–e exhibit more elliptical contour lines and steeper response surfaces than others, which demonstrates the significant interaction effects of ultrasonic power and extraction temperature, ultrasonic time and liquid/solid ratio, and ultrasonic time and extraction temperature on SMGP yields.

### 3.3. Optimized Conditions

In Design-Expert software 10.0, the optimized conditions for maximum SMGP yields were calculated as follows: ultrasonic power of 220.023 W, ultrasonic time of 1.9112 h, liquid/solid ratio of 20.842 mL/g, extraction temperature of 62.519 °C, and predicted SMGP yield of 1.640%. However, these parameters were hard to be well-controlled for industrial production; therefore, the experimental parameters were embellished as follows: ultrasonic power of 220 W, ultrasonic time of 2.0 h, liquid/solid ratio of 20 mL/g, extraction temperature of 60 °C, and SMGP yields of 1.63 ± 0.04%. Besides, in the preliminary experiment, the SMGP yields were 1.15 ± 0.06% under the initial extraction conditions (ultrasonic power of 250 W, ultrasonic time of 2.0 h, liquid/solid ratio of 20 mL/g, extraction temperature of 70 °C). These data indicated that RSM could effectively optimize the extraction parameters of SMGP, and could provide mathematical models and intuitive graphs for preferable evaluation.

### 3.4. Chemical Composition, UV, and HPGPC Analysis of SMGP

The chemical compositions of SMGP were determined using the previously mentioned methods, and the results exhibited that the average contents of polysaccharide, protein, and uronic acid in SMGP were 20.62%, 76.96%, and 1.16%, respectively, indicating the SMGP belonged to a kind of glycoprotein that employed protein as the backbone, and branched with the oligosaccharide chain.

Figure 2A displays an obvious absorption peak at 280 nm, indicating that SMGP contained large quantities of amino acids, which was consistent with the protein content determination result. As presented in Figure 2B, a uniformly distributed peak at 7.582 min was observed, suggesting that the average molecular weight of SMGP was 1.55 × 10^5^ Da based on the standard curve (*y* = 18.833 − 1.7988*x*, *r²* = 0.9991, where *y* represents the natural logarithm of molecular weight, and *x* represents the corresponding retention time).

### 3.5. Monosaccharide Compositions of SMGP

The monosaccharide compositions of SMGP were evaluated by GC, and the results are displayed in Figure 3. As presented, SMGP was primarily comprised of rhamnose (Rha), arabinose (Ara), mannose (Man), glucose (Glc), and galactose (Gal) in a molar ratio of 0.53:0.52:1.00:0.53:0.74.

### 3.6. Types and Contents of Amino Acids in SMGP

The amino acid types and contents of SMGP were determined after acid hydrolysis, and the result is shown in Table 3. As presented, there were 16 kinds of amino acids detected in SMGP, and the glutamic acid, valine, aspartic acid, and phenylalanine contents were higher than 10%, which are the typical plant-derived amino acids.

### 3.7. Antioxidant Activities In Vitro

Excessive production and accumulation of ROS can induce oxidative stress and various diseases, and severely threaten a human’s health [32]. As reported, the free radicals, including O_2_^−^, OH^−^, ABTS, and DPPH, have been widely used to evaluate the in vitro antioxidant activities of various bioactive substances. The OH^−^ and O_2_^−^ free radicals are mainly generated from mitochondrial oxidative metabolism, and are involved in regulating cellular growth and development [33]. DPPH and ABTS free radicals are stable free radicals, and have been commonly employed as substrates to evaluate the scavenging capacities of multiple compounds in vitro [34,35].

The scavenging effects of SMGP (0.25, 0.5, 1.0, 2.0, 3.0 mg/mL) on O_2_^−^, OH^−^, ABTS, and DPPH free radicals were evaluated, and the results are shown in Figure 4. The scavenging effects of SMGP on O_2_^−^, OH^−^, ABTS, and DPPH radicals were remarkably improved with the increased SMGP concentrations (from 0.25 to 3.0 mg/mL), and the maximum antioxidant capacities on O_2_^−^, OH^−^, ABTS, and DPPH free radicals reached 148.95 ± 9.98 U/L, 277.78 ± 13.89 U/mL, 89.16 ± 3.34%, and 86.82 ± 3.29%, respectively, which was significantly higher than the in vitro antioxidant effects exerted by reported individual polysaccharides or proteins [36,37]. These results demonstrated that SMGP could exhibit stronger scavenging effects on free radicals, which was consistent with the antioxidant effects of a Fupenzi glycoprotein [38].

### 3.8. Antioxidant Activities In Vivo

SOD is the only enzyme that can scavenge O_2_^−^ free radicals specifically, and SOD activity enhancement could be an effective method to improve antioxidant effects, and relieve oxidative stress [39]. GSH-Px can reduce H_2_O_2_ to H_2_O in cytosol, and this activity has been widely used to reflect the in vivo antioxidant levels [40]. The MDA can damage the structure of proteins and DNA as the final breakdown product in the lipid peroxidation process, and the concentration has been used as a positive indicator to reflect oxidative stress [41]. These antioxidant enzymes activities and MDA contents in sera were determined, and the results are displayed in Table 4.

As demonstrated, the SOD and GSH-Px activities in sera of CTX-treated mice were obviously reduced compared with the blank group, whereas the MDA contents were significantly increased, indicating that CTX could dramatically decrease the antioxidant effects in mice. However, compared with the model group, the antioxidant enzyme activities were significantly enhanced after SMGP treatments, whereas the MDA contents were obviously reduced dose-dependently, which suggested that SMGP could observably improve the antioxidant activities of mice even after CTX treatment. However, the T-AOC values in mice sera presented no significant difference among these groups, suggesting strong self-regulating capacities. As reported, *Ganoderma lucidum* polysaccharides can enhance antioxidant activities in ovarian cancer rats [42]. *Allium mongolicum* Regel polysaccharide can enhance sera’s SOD and GSH-Px activities, and decrease the contents of MDA [43,44]. In this study, the SMGP exhibited a strong enhancement of the enzyme activities in CTX-treated mice as a kind of natural edible glycoprotein, which resulted in the decrease of MDA contents in sera.

### 3.9. Thymus Indices and Immune Cell Activities

In this study, the determination results of the thymus weights/indices and immune cell activities are displayed in Figure 5. The thymus weights/indices of the model group were obviously reduced compared with that of the blank group, suggesting that CTX induced oxidative stress, and caused thymus atrophy. However, the thymus weights/indices of SMGP groups were significantly increased compared with that of the model group dose-dependently, indicating the antioxidant activities and thymus protection of SMGP in CTX-treated mice.

Figure 5C–F demonstrates the immune cell activity determination results. As shown, the immune cell activities of the model group were all observably decreased compared with that of the blank group, suggesting the immunosuppressive effects of CTX injection. However, after different concentrations of SMGP treatments, the cell activities were all dose-dependently enhanced, which demonstrated that SMGP could significantly enhance the immune cell activities in mice, even after CTX treatment.

### 3.10. Antibody Levels and Cytokine Expressions In Sera

The thymus promotes the development and differentiation of T cells, and plays a crucial role in protecting the body against immunocompromised-induced diseases, which have been employed as preliminary evaluation indicators in relevant animal experiments [26]. IL-2 and IL-4 are secreted by immune cells, and enhance T cell and NK cell activities [45,46]. TNF-α and IFN-γ exhibit direct cytotoxicity, and can also activate lymphocytes, macrophages, and NK cells to enhance immune capacity [47,48].

The expression level results of antibodies (A, IgG; B, IgM) and cytokines (C, IL-2; D, IL-4; E, TNF-α; F, IFN-γ) of each group are presented in Figure 6. As displayed, the antibody expression levels, including IgG and IgM, of mice after CTX treatment were dramatically reduced compared with that of the blank group, suggesting that CTX significantly inhibited humoral immune activity (mediated by B cells). However, SMGP treatments effectively improved the antibody secretions in a dose-dependent manner compared with the model group, indicating the immune enhancement of SMGP on B cells. Besides, the expressions of IL-2, IL-4, TNF-α, and IFN-γ of the model group were all significantly reduced compared with that of the blank group, indicating that CTX treatment could also severely suppress the immunoregulation capability in vivo. However, SMGP remarkably improved the cytokine expressions, with a dosage association, which would be of benefit to the corresponding immune cell activation for better exerting relevant immune responses.

As reported, *Ganoderma lucidum* polysaccharides could enhance the activities of multiple immune cells, and improve the expressions levels of IL-2, IFN-γ [49]. Besides, the *S. miltiorrhiza* polysaccharides have also exhibited enhancements on various leukocyte subset activities and cytokine expressions in animals [10,50]. However, the glycoprotein from *S. miltiorrhiza* and related bioactivities have rarely been researched. In this study, the SMGP was isolated by ethanol-extraction technology, and improved the immune-associated cytokine/antibody expressions in CTX-treated mice, thereby effectively enhancing the corresponding immune cell activities.

## 4. Conclusions

The optimal extraction parameters for SMGP were identified as follows: ultrasonic power of 220 W, ultrasonic time of 2.0 h, liquid/solid ratio of 20 mL/g, extraction temperature of 60 °C, and yields of 1.63 ± 0.04%. The prepared SMGP was mainly composed of protein and polysaccharide (contents of 76.96% and 20.62%, respectively), with the average molecular weight of 1.55 × 10^5^ Da. Moreover, SMGP presented superior scavenging effects on free radicals in vitro, and exhibited strong antioxidant activities and immunological enhancement on various immune cells in CTX-treated mice. These data could provide a new extraction technology for water-soluble active substances, and the theoretical foundation for the practical application of SMGP as an antioxidant or immunopotentiator in the food industry.

## Figures and Tables

**Figure 1 foods-11-00705-f001:**
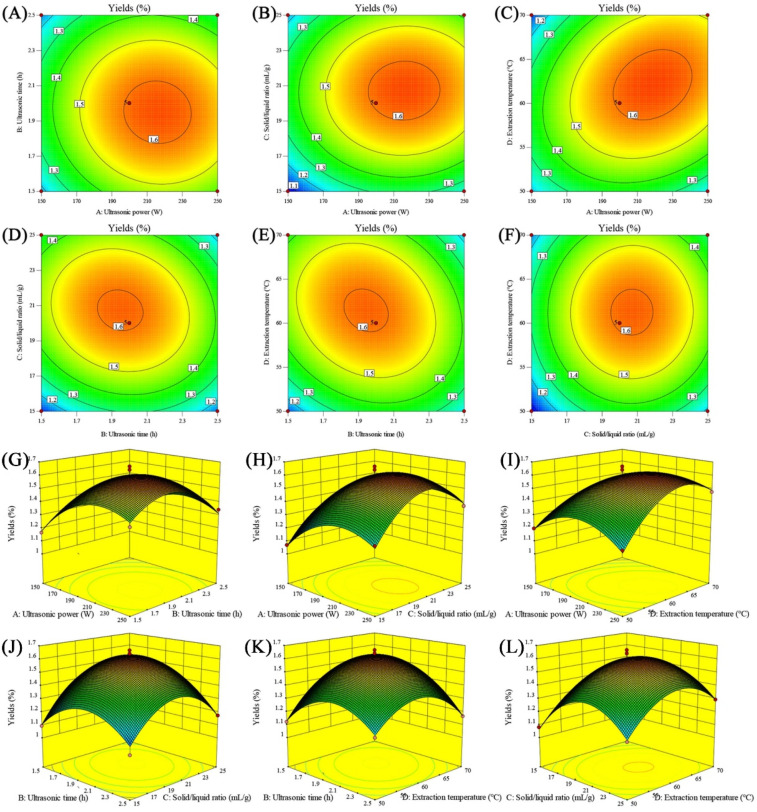
Two-dimensional and three-dimensional maps. (**A**,**G**): ultrasonic power and ultrasonic time; (**B**,**H**): ultrasonic power and liquid/solid ratio; (**C**,**I**): ultrasonic power and extraction temperature; (**D**,**J**): ultrasonic time and liquid/solid ratio; (**E**,**K**): ultrasonic time and extraction temperature; (**F**,**L**): liquid/solid ratio and extraction temperature.

**Figure 2 foods-11-00705-f002:**
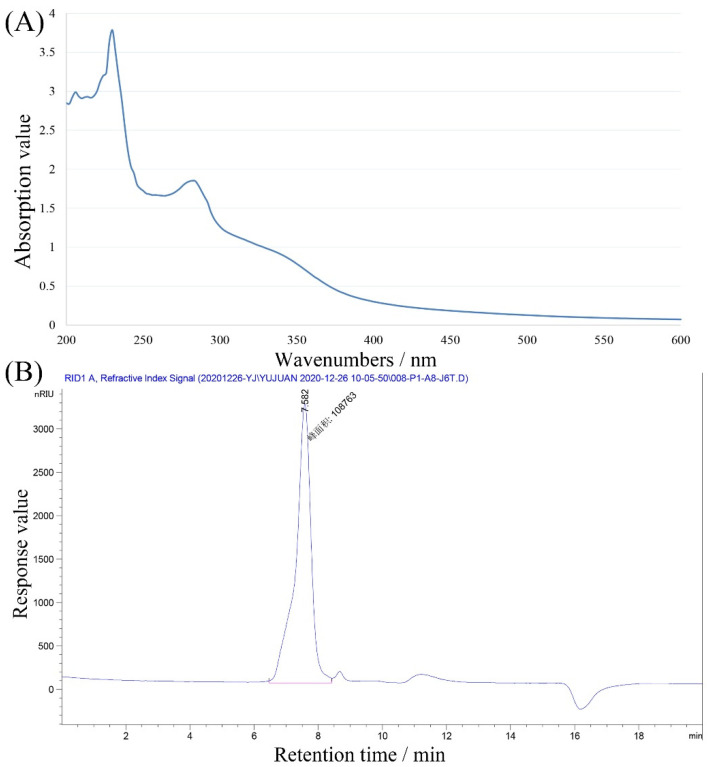
UV (**A**) and HPGPC (**B**) spectra of SMGP.

**Figure 3 foods-11-00705-f003:**
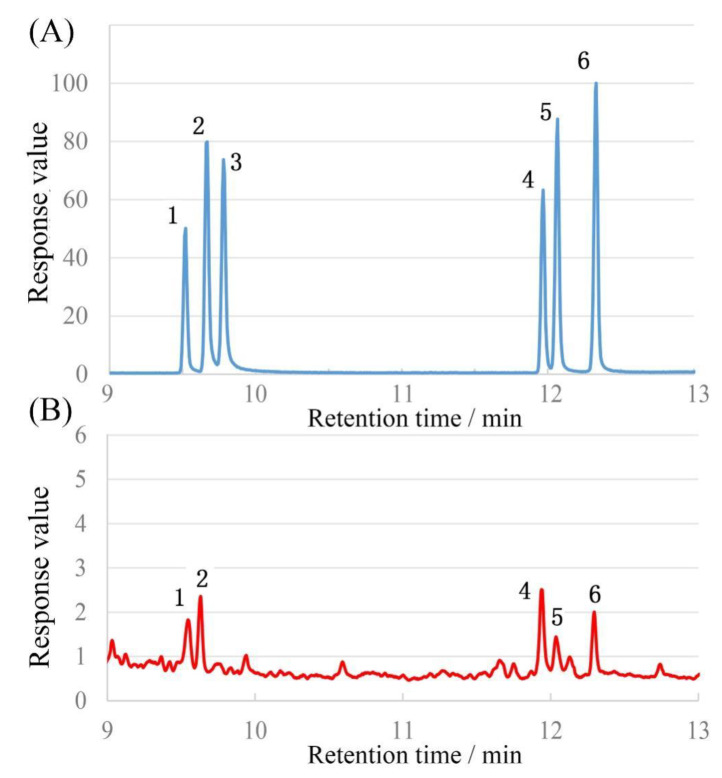
Monosaccharide composition results of SMGP. (**A**), monosaccharide standards; (**B**), SMGP derivatives. Note: 1—L-rhamnose; 2—D-arabinose; 3—D-xylose; 4—D-mannose; 5—D-glucose; 6—D-galactose.

**Figure 4 foods-11-00705-f004:**
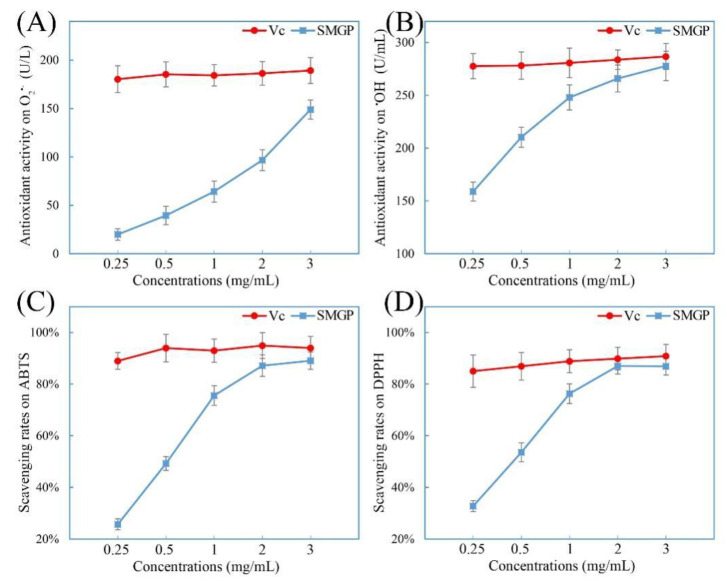
Free radical scavenging capacities ((**A**) O_2_^−^; (**B**) OH^−^; (**C**) ABTS; (**D**) DPPH) of SMGP.

**Figure 5 foods-11-00705-f005:**
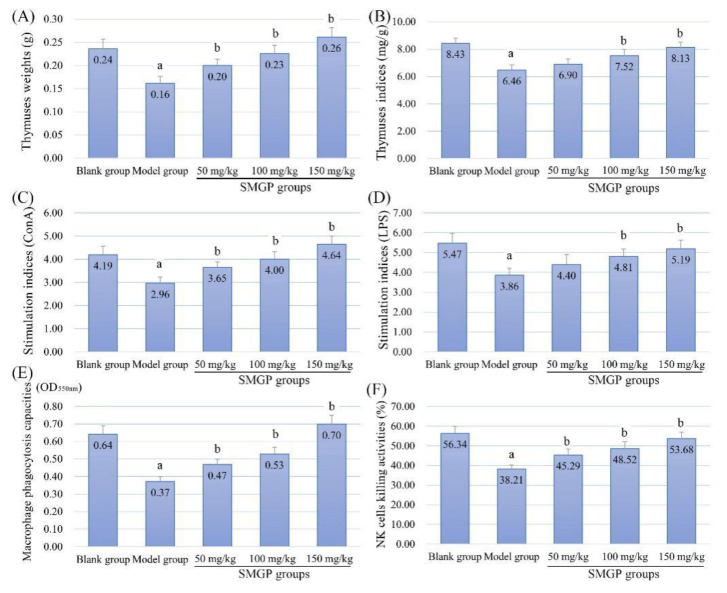
Thymus indicators ((**A**) weights; (**B**) indices) and immune cell activities ((**C**) T lymphocytes proliferation ability; (**D**) B lymphocytes proliferation ability; (**E**) macrophages phagocytosis capacity; (**F**) NK cell killing activities) of mice. Note: a, *p* < 0.05 compared with blank group; b, *p* < 0.05 compared with model group.

**Figure 6 foods-11-00705-f006:**
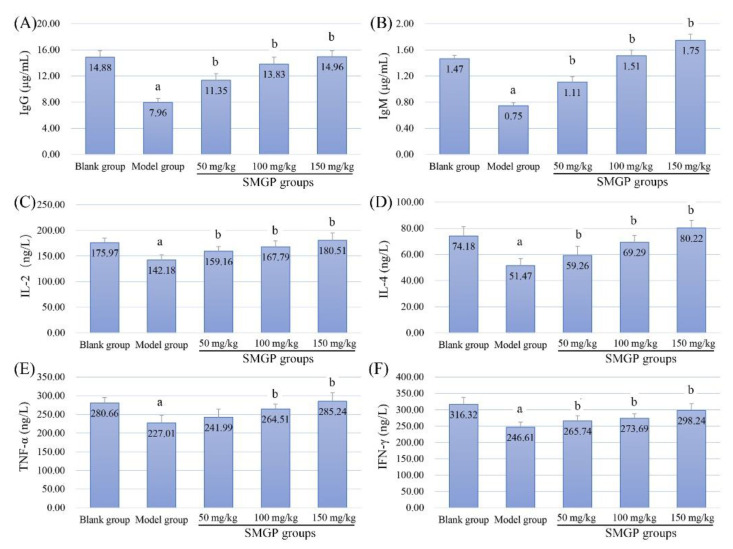
Antibody levels ((**A**) IgG; (**B**) IgM) and cytokines expressions ((**C**) IL-2; (**D**) IL-4; (**E**) TNF-α; (**F**) IFN-γ) of mice in each group. Note: a, *p* < 0.05 compared with blank group; b, *p* < 0.05 compared with model group.

**Table 1 foods-11-00705-t001:** Actual and predicted SMGP yields.

	A	B	C	D	Yields	
Run	Ultrasonic Power	Ultrasonic Time	Solid/Liquid Ratio	Extraction Temperature	Actual Value	Predicted Value
	W	h	mL/g	°C	%	%
1	150	1.5	20	60	1.17	1.18
2	250	1.5	20	60	1.38	1.41
3	150	2.5	20	60	1.2	1.19
4	250	2.5	20	60	1.34	1.31
5	200	2	15	50	1.08	1.06
6	200	2	25	50	1.17	1.17
7	200	2	15	70	1.18	1.15
8	200	2	25	70	1.29	1.29
9	150	2	20	50	1.2	1.22
10	250	2	20	50	1.22	1.21
11	150	2	20	70	1.08	1.14
12	250	2	20	70	1.48	1.49
13	200	1.5	15	60	1.09	1.1
14	200	2.5	15	60	1.08	1.14
15	200	1.5	25	60	1.39	1.33
16	200	2.5	25	60	1.17	1.18
17	150	2	15	60	1.07	1.06
18	250	2	15	60	1.25	1.24
19	150	2	25	60	1.14	1.17
20	250	2	25	60	1.37	1.39
21	200	1.5	20	50	1.12	1.13
22	200	2.5	20	50	1.2	1.19
23	200	1.5	20	70	1.38	1.38
24	200	2.5	20	70	1.16	1.17
25	200	2	20	60	1.59	1.61
26	200	2	20	60	1.67	1.61
27	200	2	20	60	1.64	1.61
28	200	2	20	60	1.57	1.61
29	200	2	20	60	1.56	1.61

**Table 2 foods-11-00705-t002:** Analysis of variance for quadratic model.

Sources	Sum of	Degree of Freedom	Mean	*F*	*p*-Value	Significance
	Squares		Square	Value	Prob > *F*	
Model	0.92	14	0.066	39.34	<0.0001	Significant
*A*	0.12	1	0.12	70.08	<0.0001	
*B*	0.012	1	0.012	7.07	0.0187	
*C*	0.049	1	0.049	29.49	<0.0001	
*D*	0.029	1	0.029	17.63	0.0009	
*AB*	1.27 × 10^−3^	1	1.27 × 10^−3^	0.76	0.3973	
*AC*	5.02 × 10^−4^	1	5.02 × 10^−4^	0.3	0.5924	
*AD*	0.033	1	0.033	19.81	0.0005	
*BC*	0.01	1	0.01	6.08	0.0272	
*BD*	0.022	1	0.022	12.96	0.0029	
*CD*	1.04 × 10^−4^	1	1.04 × 10^−4^	0.062	0.8066	
*A^2^*	0.16	1	0.16	97.13	<0.0001	
*B^2^*	0.22	1	0.22	132.53	<0.0001	
*C^2^*	0.36	1	0.36	212.71	<0.0001	
*D^2^*	0.26	1	0.26	152.64	<0.0001	
Residual	0.023	14	1.67 × 10^−3^			
Lack of Fit	0.015	10	1.52 × 10^−3^	0.75	0.6792	not significant
Pure Error	8.17 × 10^−3^	4	2.04 × 10^−3^			
Cor Total	0.94	28				
*R^2^*	0.9752		Adj *R^2^*	0.9504		
Pred *R^2^*	0.8935		C.V. %	3.18		

**Table 3 foods-11-00705-t003:** Types and contents of amino acids in SMGP.

Amino Acids	Abbreviations	Contents (%)	Amino Acids	Abbreviations	Contents (%)
Glutamic acid	Glu	12.97%	Leucine	Leu	5.08%
Valine	Val	12.03%	Isoleucine	Ile	4.89%
Aspartic acid	Asp	10.71%	Glycine	Gly	4.32%
Phenylalanine	Phe	10.34%	Alanine	Ala	3.95%
Threonine	Thr	8.83%	Arginine	Arg	3.01%
Serine	Ser	6.77%	Lysine	Lys	2.63%
Proline	Pro	6.20%	Histidine	His	1.50%
Tyrosine	Tyr	5.64%	Methionine	Met	1.13%

**Table 4 foods-11-00705-t004:** Antioxidant enzyme activities and MDA contents.

Groups		SOD (Unit)	GSH-Px (Unit)	MDA (nmol)	T-AOC (Unit)
Blank group		148.53 ± 8.35	355.47 ± 24.37	4.39 ± 0.37	34.24 ± 2.15
Model group		124.26 ± 9.26 ^a^	298.38 ± 18.42 ^a^	6.69 ± 0.47 ^a^	33.86 ± 1.87
SMGP groups	50 mg/kg	139.48 ± 7.18 ^b^	325.56 ± 19.49 ^b^	5.63± 0.49 ^b^	35.02 ± 1.49
	100 mg/kg	153.37 ± 9.05 ^b^	342.48 ± 21.47 ^b^	5.03 ± 0.32 ^b^	33.84 ± 2.16
	150 mg/kg	166.58 ± 8.79 ^b^	367.16 ± 22.21 ^b^	4.17 ± 0.33 ^b^	34.93 ± 1.24

Note: ^a^, *p* < 0.05 compared with blank group; ^b^
*p* < 0.05 compared with model group.

## Data Availability

Data is contained within the article.

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
