# Peer review of "Preparation, Antioxidant and Immunoregulatory Activities of a Macromolecular Glycoprotein from Salvia miltiorrhiza"

_foods, 2022, doi:10.3390/foods11050705_

Round 1

Reviewer 1 Report

The manuscript under appreciation is about the optimization of the ultrasonic-assisted extraction of proteoglycan from Salvia miltiorrhiza using Box-Behnken design-response surface methodology. Structural identification and determination of bioactivity were also performed.

The manuscript is interesting and provides novelty. However, there are several points that must be addressed and the explanations should be added to the manuscript.

Introduction: The authors' isolated proteoglycan from the roots. However, is not clear whether the proteoglycan generally exists at the roots of the plants and/or in other parts of the plant. Therefore a relevant phrase must be added. Furthermore, in the last paragraph, the authors must provide more information about the novelty of this study.

Subsection 2.2: Please provide information about the ultrasonic equipment used (model, vendor).

Subsection 2.3: Regarding the application of Box-Behnken design, why did the authors choose the specific design over the Central Composite Design?

Table 1: Please provide more details on the determination and calculation of SMPG yield which was used as a response.

Section 3: add a discussion based on Table 2 about which factors had the most significant contribution in the extraction yield of SMPG and make a comparison with other literature studies about the importance of these factors. Also, provide an explanation of why have you chosen the quadratic model. Also, the discussion on the analysis of the interactive effects is poor. Please add further comments about what we observe at the contour plots.

Author Response

Reviewer #1:

Comments and Suggestions for Authors

Q1: The manuscript under appreciation is about the optimization of the ultrasonic-assisted extraction of proteoglycan from Salvia miltiorrhiza using Box-Behnken design-response surface methodology. Structural identification and determination of bioactivity were also performed.

The manuscript is interesting and provides novelty. However, there are several points that must be addressed and the explanations should be added to the manuscript.

Response: Thank you very much for the kind evaluation, the following comments have been answered and the manuscript has been modified under revisions mode.

Q2: Introduction: The authors' isolated proteoglycan from the roots. However, is not clear whether the proteoglycan generally exists at the roots of the plants and/or in other parts of the plant. Therefore, a relevant phrase must be added. Furthermore, in the last paragraph, the authors must provide more information about the novelty of this study.

Response: The relevant expressions have been improved for explaining the compounds location in plants, and for better demonstrating the novelty and significance of this study.

Q3: Subsection 2.2: Please provide information about the ultrasonic equipment used (model, vendor).

Response: The relevant information has been supplemented.

Q4: Subsection 2.3: Regarding the application of Box-Behnken design, why did the authors choose the specific design over the Central Composite Design?

Response: As is known to us, Box-Behnken Design could provide an equation of these factors and employ graphic technology for intuitive interaction effects of various combinations, and would be more benefit for selecting the optimal conditions under less parameters (< 5) compared with Central Composite Design. Therefore, Box-Behnken Design was applied in this study for the extraction optimization in this study after referring to many relevant reported researches.

Q5: Table 1: Please provide more details on the determination and calculation of SMPG yield which was used as a response.

Response: The relevant expressions have been supplemented.

Q6: Section 3: add a discussion based on Table 2 about which factors had the most significant contribution in the extraction yield of SMPG and make a comparison with other literature studies about the importance of these factors. Also, provide an explanation of why have you chosen the quadratic model. Also, the discussion on the analysis of the interactive effects is poor. Please add further comments about what we observe at the contour plots.

Response: The relevant discussion has been supplemented in the related section.

Finally, Thank you very much for all these kind suggestions. We sincerely hope that these revisions would improve the quality of this paper and solve your concerns.

Reviewer 2 Report

In this study, the authors presented the optimized extraction conditions for proteoglycan (SMPG) from the roots of Salvia miltiorrhiza (via response surface methodology), structural properties of SMPG, as well as its antioxidant and immunoregulatory potential.

Strength of the study is that it has used diverse methodologies to characterize SMPG, including chemical assays and animal models. The abstract was nicely written, highlighting key findings in the study. Furthermore, rationales in experimental design, e.g., use of 60% ethanol in extraction (lines 53-58), or minor adjustment of optimized conditions (lines 187-190) considering practicality in industrial application, were carefully explained. Overall, it is a well-written and easy-to-read manuscript.

I found no major issues/flaws in the paper.

Below are a few very minor feedbacks I have for the authors’ consideration.

  1. Section 2.12 - Statistical analysis: Were all the measurements, e.g., antioxidant assays, done in triplicates? If so, it would be good to indicate the number of replicates here.

  1. Line 147: Please introduce the abbreviation “NK” in full at the first mention.

  1. Fig 2(A): Besides 280 nm, there is another peak between 200 nm and 250 nm. What can be deduced from this first peak on the left – which seems more prominent than the 280 nm peak?

  1. Lines 215-216: Are the percentage ranges reported in Table 3 similar to what have been reported in other studies on this or similar samples in the literature?

  1. Lines 233-235: “…SMPG could exhibit stronger scavenging effects on free radicals, which might be caused by the linkages between amino acids and polysaccharides.” – This suggestion seems somewhat unclear. Could the authors please briefly elaborate how the “linkages” could boost the radical scavenging activities of SMPG? Are there any other reports in the literature that can be cited to support this suggestion?

Author Response

Reviewer #2:

Comments and Suggestions for Authors

Q1: In this study, the authors presented the optimized extraction conditions for proteoglycan (SMPG) from the roots of Salvia miltiorrhiza (via response surface methodology), structural properties of SMPG, as well as its antioxidant and immunoregulatory potential.

Strength of the study is that it has used diverse methodologies to characterize SMPG, including chemical assays and animal models. The abstract was nicely written, highlighting key findings in the study. Furthermore, rationales in experimental design, e.g., use of 60% ethanol in extraction (lines 53-58), or minor adjustment of optimized conditions (lines 187-190) considering practicality in industrial application, were carefully explained. Overall, it is a well-written and easy-to-read manuscript.

I found no major issues/flaws in the paper.

Response: Thank you very much for these valuable comments, the following questions have been answered and the manuscript has been modified under revisions mode.

Below are a few very minor feedbacks I have for the authors’ consideration.

Q2: Section 2.12 - Statistical analysis: Were all the measurements, e.g., antioxidant assays, done in triplicates? If so, it would be good to indicate the number of replicates here.

Response: Thank you for the kind suggestion, the relevant expression has been supplemented.

Q3: Line 147: Please introduce the abbreviation “NK” in full at the first mention.

Response: The full name of “NK” has been introduced at the first appearance.

Q4: Fig 2(A): Besides 280 nm, there is another peak between 200 nm and 250 nm. What can be deduced from this first peak on the left – which seems more prominent than the 280 nm peak?

Response: The peaks between 200 nm and 250 nm were attributed to various compounds including amino acids and polysaccharides, while peak at 280 nm was always used as an indicator to specifically reflect the existence of protein.

Q5: Lines 215-216: Are the percentage ranges reported in Table 3 similar to what have been reported in other studies on this or similar samples in the literature?

Response: The percentage ranges reported in Table 3 were similar with the following paper in Chinese journal (胡建林,张荣平,李惠兰,曹树明,杨丽川.大紫丹参中氨基酸和微量元素的含量分析[J].中国中医药科技,2002(03):166.), but we could not find a SCI paper involving Salvia miltiorrhiza extracted protein and the amino acids compositions, thus there was no reference cited in the section.

Q6: Lines 233-235: “…SMPG could exhibit stronger scavenging effects on free radicals, which might be caused by the linkages between amino acids and polysaccharides.” – This suggestion seems somewhat unclear. Could the authors please briefly elaborate how the “linkages” could boost the radical scavenging activities of SMPG? Are there any other reports in the literature that can be cited to support this suggestion?

Response: The relevant expressions have been modified in the related section, and a proper reference has been cited.

Finally, thank you very much for all these constructive comments, we hope that these revisions would solve your main concerns and obviously improve the qualities of this manuscript.